# Mastering Atari Games with Limited Data

**Weirui Ye**[*]   **Shaohuai Liu**[*]   **Thanard Kurutach**[†]   **Pieter Abbeel**[†]   **Yang Gao**[*‡]
[*]Tsinghua University, [†]UC Berkeley, [‡] Shanghai Qi Zhi Institute

## Abstract

Reinforcement learning has achieved great success in many applications. However, sample efficiency remains a key challenge, with prominent methods requiring millions (or even billions) of environment steps to train. Recently, there has been significant progress in sample efficient image-based RL algorithms; however, consistent human-level performance on the Atari game benchmark remains an elusive goal. We propose a sample efficient model-based visual RL algorithm built on MuZero, which we name EfficientZero. Our method achieves 190.4% mean human performance and 116.0% median performance on the Atari 100k benchmark with only two hours of real-time game experience and outperforms the state SAC in some tasks on the DMControl 100k benchmark. This is the first time an algorithm achieves super-human performance on Atari games with such little data. EfficientZero's performance is also close to DQN's performance at 200 million frames while we consume 500 times less data. EfficientZero's low sample complexity *and* high performance can bring RL closer to real-world applicability. We implement our algorithm in an easy-to-understand manner and it is available at https://github.com/YeWR/EfficientZero. We hope it will accelerate the research of MCTS-based RL algorithms in the wider community.

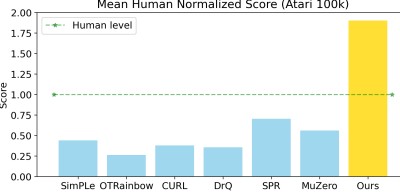
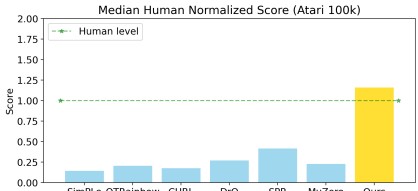

Figure 1: Our proposed method EfficientZero is 170% and 180% better than the previous SoTA performance in mean and median human normalized score and is the first to outperform the average human performance on the Atari 100k benchmark. The high sample efficiency and performance of EfficientZero can bring RL closer to the real-world applications.

## 1 Introduction

Reinforcement learning has achieved great success on many challenging problems. Notable work includes DQN [24], AlphaGo [33] and OpenAI Five [5]. However, most of these works come at the cost of a large number of environmental interactions. For example, AlphaZero [34] needs to play 21 million games at training time. On the contrary, a professional human player can only play around 5 games per day, meaning it would take a human player 11,500 years to achieve the same amount of experience. The sample complexity might be less of an issue when applying RL algorithms in simulation and games. However, when it comes to real-world problems, such as robotic

---

[*]{ywr20, liush20}@mails.tsinghua.edu.cn, gaoyangiiis@tsinghua.edu.cn
[†]{thanard.kurutach, pabbeel}@berkeley.edu

35th Conference on Neural Information Processing Systems (NeurIPS 2021).

manipulation, healthcare, and advertisement recommendation systems, achieving high performance while maintaining low sample complexity is the key to viability.

People have made a lot of progress in sample efficient RL in the past years [8, 10, 35, 22, 21, 32, 18]. Among them, model-based methods have attracted a lot of attention, since both the data from real environments and the "imagined data" from the model can be used to train the policy, making these methods particularly sample-efficient [8, 10]. However, most of the successes are in state-based environments. In image-based environments, some model-based methods such as MuZero [27] and Dreamer V2 [14] achieve super-human performance, but they are not sample efficient; other methods such as SimPLe [18] is quite efficient but achieve inferior performance (0.144 human normalized median scores). Recently, data-augmented and self-supervised methods applied to model-free methods have achieved more success in the data-efficient regime [32]. However, they still fail to achieve the levels which can be expected of a human.

Therefore, for improving the sample efficiency as well as keeping superior performance, we find the following three components are essential to the model-based visual RL agent: a self-supervised environment model, a mechanism to alleviate the model compounding error, and a method to correct the off-policy issue. In this work, we propose EfficientZero, a model-based RL algorithm that achieves high performance with limited data. Our proposed method is built on MuZero. We make three critical changes: (1) use self-supervised learning to learn a temporally consistent environment model, (2) learn the *value prefix* in an end-to-end manner, thus helping to alleviate the compounding error in the model, (3) use the learned model to correct off-policy value targets.

As illustrated as Figure 1, our model achieves state-of-the-art performance on the widely used Atari [4] 100k benchmark and it achieves super-human performance with only 2 hours of real-time gameplay. More specifically, our model achieves 190.4% mean human normalized performance and 116.0% median human normalized performance. As a reference, DQN [24] achieves 220% mean human normalized performance, and 96% median human normalized performance, at the cost of 500 times more data (200 million frames). To further verify the effectiveness of EfficientZero, we conduct experiments on some simulated robotics environments of the DeepMind Control (DMControl) suite. It achieves state-of-the-art performance and outperforms the state SAC which directly learns from the ground truth states. Our sample efficient and high-performance algorithm opens the possibility of having more impact on many real-world problems.

## 2 Related Work

### 2.1 Sample Efficient Reinforcement Learning

Sample efficiency has attracted significant work in the past. In RL with image inputs, model-based approaches [13, 12] which model the world with both a stochastic and a deterministic component, have achieved promising results for simulated robotic control. Kaiser et al. [18] propose to use an action-conditioned video prediction model, along with a policy learning algorithm. It achieves the first strong performance on Atari games with as little as 400k frames. However, Kielak [19] and van Hasselt et al. [39] argue that this is not necessary to achieve strong results with model-based methods, and they show that when tuned appropriately, Rainbow [16] can achieve comparable results.

Recent advances in self-supervised learning, such as SimCLR [6], MoCo [15], SimSiam [7] and BYOL [11] have inspired representation learning in image-based RL. Srinivas et al. [35] propose to use contrastive learning in RL algorithms and their work achieves strong performance on image-based continuous and discrete control tasks. Later, Laskin et al. [22] and Kostrikov et al. [21] find that contrastive learning is not necessary, but with data augmentations alone, they can achieve better performance. Schwarzer et al. [32] propose a temporal consistency loss, which is combined with data augmentations and achieves state-of-the-art performance. Notably, our self-supervised consistency loss is quite similar to Schwarzer et al. [32], except we use SimSiam [6] while they use BYOL [11] as the base self-supervised learning framework. However, Schwarzer et al. [32] only apply the learned representations in a model-free manner, while we combine the learned model with model-based exploration and policy improvement, thus leading to more efficient use of the environment model.

Despite the recent progress in the sample-efficient RL, today's RL algorithms are still well behind human performance when the amount of data is limited. Although traditional model-based RL is considered more sample efficient than model-free ones, current model-free methods dominate in

terms of performance for image-input settings. In this paper, we propose a model-based RL algorithm that for the first time, achieves super-human performance on Atari games with limited data.

## 2.2 Reinforcement Learning with MCTS

Temporal difference learning [24, 38, 40, 16] and policy gradient based methods [25, 23, 29, 31] are two types of popular reinforcement learning algorithms. Recently, Silver et al. [33] propose to use MCTS as a policy improvement operator and has achieved great success in many board games, such as Go, Chess, and Shogi [34]. Later, the algorithm is adapted to learn the world model at the same time [27]. It has also been extended to deal with continuous action spaces [17] and offline data [28]. These MCTS RL algorithms are a hybrid of model-based learning and model-free learning.

However, most of them are trained with a lot of environmental samples. Our method is built on top of MuZero [27], and we demonstrate that our method can achieve higher sample efficiency while still achieving competitive performance on the Atari 100k benchmark. de Vries et al. [9] have studied the potential of using auxiliary loss similar to our self-supervised consistency loss. However, they only test on two low dimensional state-based environments and find the auxiliary loss has mixed effects on the performance. On the contrary, we find that the consistency loss is critical in most environments with high dimensional observations and limited data.

## 2.3 Multi-Step Value Estimation

In Q-learning [41], the target Q value is computed by one step backup. In practice, people find that incorporating multiple steps of rewards at once, i.e. $z_t = \sum_{i=0}^{k-1} \gamma^i u_{t+i} + \gamma^k v_{t+k}$, where $u_{t+i}$ is the reward from the replay buffer, $v_{t+k}$ is the value estimation from the target network, to compute the value target $z_t$ leads to faster convergence [24, 16]. However, the use of multi-step value has off-policy issues, since $u_{t+i}$ are not generated by the current policy. In practice, this issue is usually ignored when there is a large amount of data since the data can be thought as approximately on-policy. TD($\lambda$) [36] and GAE [30] improve the value estimation by better trading off the bias and the variance, but they do not deal with the off-policy issue. Recently, image input model-based algorithms such as Kaiser et al. [18] and Hafner et al. [12] use model imaginary rollouts to avoid the off-policy issue. However, this approach has the risk of model exploitation. Asadi et al. [2] proposed a multi-step model to combat the compounding error. Our proposed model-based off-policy correction method starts from the rewards in the real-world experience and uses model-based value estimate to bootstrap. Our approach balances between the off-policy issue and model exploitation.

## 3 Background

### 3.1 MuZero

Our method is built on top of the MuZero Reanalyze [27] algorithm. For brevity, we refer to it as MuZero throughout the paper. MuZero is a policy learning method based on the Monte-Carlo Tree Search (MCTS) algorithm. The MCTS algorithm operates with an environment model, a prior policy function, and a value function. The environment model is represented as the reward function $\mathcal{R}$ and the dynamic function $\mathcal{G}$: $r_t = \mathcal{R}(s_t, a_t)$, $\hat{s}_{t+1} = \mathcal{G}(s_t, a_t)$, which are needed when MCTS expands a new node. In MuZero, the environment model is learned. Thus the reward and the next state are approximated. Besides, the predicted policy $p_t = $ acts as a search prior over actions of a node. It helps the MCTS focus on more promising actions when expanding the node. MCTS also needs a value function $\mathcal{V}(s_t)$ that measures the expected return of the node $s_t$, which provides a long-term evaluation of the tree's leaf node without further search. MCTS will output an action visit distribution $\pi_t$ over the root node, which is potentially a better policy, compared to the current neural network. Thus, the MCTS algorithm can be thought of as a policy improvement operator.

In practice, the environment model, policy function, and value function operate on a hidden abstract state $s_t$, both for computational efficiency and ease of environment modeling. The abstract state is extracted by a representation function $\mathcal{H}$ on observations $o_t$: $s_t = \mathcal{H}(o_t)$. All of the mentioned models above are usually represented as neural networks. During training, the algorithm collects roll-out data in the environment using MCTS, resulting in potentially higher quality data than the current neural network policy. The data is stored in a replay buffer. The optimizer minimizes the

following loss on the data sampled from the replay buffer:

$$\mathcal{L}(u_t, r_t) + \lambda_1 \mathcal{L}(\pi_t, p_t) + \lambda_2 \mathcal{L}(z_t, v_t) \tag{1}$$

Here, $u_t$ is the reward from the environment, $r_t = \mathcal{R}(s_t, a_t)$ is the predicted reward, $\pi_t$ is the output visit count distribution of the MCTS, $p_t = \mathcal{P}(s_t)$ is the predicted policy, $z_t = \sum_{i=0}^{k-1} \gamma^i u_{t+i} + \gamma^k v_{t+k}$ is the bootstrapped value target and $v_t = \mathcal{V}(s_t)$ is the predicted value. Specifically, the reward function $\mathcal{R}$, policy function $\mathcal{P}$, value function $\mathcal{V}$, the representation function $\mathcal{H}$ and the dynamics function $\mathcal{G}$ are trainable neural networks. It is worth noting that MuZero does not explicitly learn the environment model. Instead, it solely relies on the reward, value, and policy prediction to learn the model.

## 3.2 Monte-Carlo Tree Search

Monte-Carlo Tree Search [1, 33, 34, 14], or MCTS, is a heuristic search algorithm. In our setup, MCTS is used to find an action policy that is better than the current neural network policy.

More specifically, MCTS needs an environment model, including the reward function and the next-state function. It also needs a value function and a policy function, which act as heuristics for the tree search. MCTS operates by expanding a search tree from the current node. It saves computation by selectively expanding a few nodes. In order to find a high-quality decision, the tree expansion process has to balance between exploration versus exploitation, i.e. balance between expanding a node that is promising with many visits versus expanding a node with lower performance but fewer visits. MCTS employs the UCT [26, 20] rule, i.e. UCB [3] on trees. At every node expansion step, UCT will select a node as follows [14]:

$$a^k = \arg\max_a \left\{ Q(s,a) + P(s,a) \frac{\sqrt{\sum_b N(s,b)}}{1 + N(s,a)} \left( c_1 + \log\left( \frac{\sum_b N(s,b) + c_2 + 1}{c_2} \right) \right) \right\} \tag{2}$$

where, $Q(s,a)$ is the current estimate of the Q-value, $P(s,a)$ is the current neural network policy for selecting this action, helping the MCTS prioritize exploring promising part of the tree. During training time, $P(s,a)$ is usually perturbed by noises to allow explorations. $N(s,a)$ denotes how many times this state-action pair is visited in the tree search, and $N(s,b)$ denote that of $a$'s siblings. Thus this term will encourage the search to visit the nodes whose siblings are visited often, but itself less visited. Finally, the last term gives a weights to the previous terms.

After expanding the nodes for a pre-defined number of times, the MCTS will return how many times each action under the root node is visited, as the improved policy to the root node. Thus, MCTS can be considered as a policy improvement operator in the RL setting.

## 4 EfficientZero

Model-based algorithms have achieved great success in sample-efficient learning from low-dimensional states. However, current visual model-based algorithms either require large amounts of training data or exhibit inferior performance to model-free algorithms in data-limited settings [32]. Many previous works even suspect whether model-based algorithms can really offer data efficiency when using image observations [39]. We provide a positive answer here. We propose the EfficientZero, a model-based algorithm built on the MCTS, that achieves super-human performance on the 100k Atari benchmark, outperforming the previous SoTA to a large degree.

When directly running MCTS-based RL algorithms such as MuZero, we find that they do not perform well on the limited-data benchmark. Through our ablations, we confirm the following three issues which pose challenges to algorithms like MuZero in data-limited settings.

**Lack of supervision on environment model**. First, the learned model in the environment dynamics is only trained through the reward, value and policy functions. However, the reward is only a scalar signal and in many scenarios, the reward will be sparse. Value functions are trained with bootstrapping, and thus are noisy. Policy functions are trained with the search process. None of the reward, value and policy losses can provide enough training signals to learn the environment model.

**Hardness to deal with aleatoric uncertainty**. Second, we find that even with enough data, the predicted rewards still have large prediction errors. This is caused by the aleatoric uncertainty of the underlying environment. For example, the environment is hard to model. The reward prediction

errors will accumulate when expanding the MCTS tree to a large depth, resulting in sub-optimal performance in exploration and evaluation.

**Off-policy issues of multi-step value**. Lastly, when computing the value target, MuZero uses the multi-step reward observed in the environment. Although this allows the reward to be propagated to the value function faster, we find that it suffers from severe off-policy issues and hinders convergence in the limited data scenario.

To address the above issues, we propose the following three critical modifications, which can greatly improve performance when samples are limited.

### 4.1 Self-Supervised Consistency Loss

In previous MCTS RL algorithms, the environment model is either given or only trained with rewards, values, and policies, which cannot provide sufficient training signals due to their scalar nature. The problem is more severe when the reward is sparse or the bootstrapped value is not accurate. The MCTS policy improvement operator heavily relies on the environment model. Thus, it is vital to have an accurate one.

We notice that the output $\hat{s}_{t+1}$ from the dynamic function $\mathcal{G}$ should be the same as $s_{t+1}$, i.e. the output of the representation function $\mathcal{H}$ with input of the next observation $o_{t+1}$ (Fig. 2). This can help to supervise the predicted next state $\hat{s}_{t+1}$ using the actual $s_{t+1}$, which is a tensor with at least a few hundred dimensions. This provides $\hat{s}_{t+1}$ with much more training signals than the default scalar reward and value.

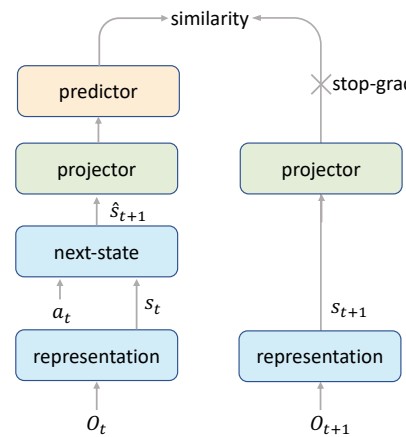

Figure 2: The self-supervised consistency loss.

More specifically, we adopt the recently proposed Sim-Siam [7] self-supervised framework. SimSiam [7] is a self-supervised method that takes two augmentation views of the same image and pulls the output of the second branch close to that of the first branch, where the first branch is an encoder network without gradient, and the second branch is the same encoder network with the gradient and a predictor head. The predictor head can simply be a two-layer MLP.

Note that SimSiam only learns the representation of individual images, and is not aware of how different images are connected. The learned image representations of SimSiam might not be a good candidate for learning the environment transition function, since adjacent observations might be encoded to very different representation encodings. We propose a self-supervised method that learns the transition function, along with the image representation function in an end-to-end manner. Figure 2 shows our method. Since we aim to learn the transition between adjacent observations, we pull $o_t$ and $o_{t+1}$ close to each other. The transition function is applied after the representation of $o_t$, such that $s_t$ is transformed to $\hat{s}_{t+1}$, which now represents the same entity as the other branch. Then both of $s_{t+1}$ and $\hat{s}_{t+1}$ go through a common projector network. Since $s_{t+1}$ is potentially a more accurate description of $o_{t+1}$ compared to $\hat{s}_{t+1}$, we make the $o_{t+1}$ branch as the target branch. It is common in self-supervised learning that the second or the third layer from the last is chosen as the features for some reason. Here, we choose the outputs from the representation network or the dynamics network as the hidden states rather than those from the projector or the predictor.

The two adjacent observations provide two views of the same entity. In practice, we find that applying augmentations to observations such as a random small shift of 0-4 pixels on the image helps to further improve the learned representation quality [35, 32]. We also unroll the dynamic function recurrently for 5 further steps and also pull $\hat{s}_{t+k}$ close to $s_{t+k}$ ($k = 1, ..., 5$). Please see the Appendix for more implementation details.

### 4.2 End-To-End Prediction of the Value Prefix

In model-based learning, the agent needs to predict the future states conditioned on the current state and a series of hypothetical actions. The longer the prediction, the harder to predict it accurately, due

to the compounding error in the recurrent rollouts. This is called the state aliasing problem. The environment model plays an important role in MCTS. The state aliasing problem harms the MCTS expansion, which will result in sub-optimal exploration as well as sub-optimal action search.

Predicting the reward from an aliased state is a hard problem. For example, as shown in Figure 3, the right agent loses the ball. If we only see the first observation, along with future actions, it is very hard both for an agent and a human to predict at which exact future timestep the player would lose a point. However, it is easy to predict the agent will miss the ball after a sufficient number of timesteps if he does not

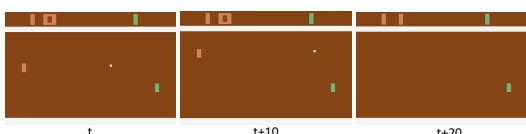

Figure 3: A sample trajectory from the Atari Pong game. In this case, the right player didn't move and missed the ball.

move. In practice, a human will never try to predict the exact step that he loses the point but will imagine over a longer horizon and thus get a more confident prediction.

Inspired by this intuition, we propose an end-to-end method to predict the *value prefix*. We notice that the predicted reward is always used in the estimation of the Q-value $Q(s, a)$ in UCT of Equation 2

$$Q(s_t, a) = \sum_{i=0}^{k-1} \gamma^i r_{t+i} + \gamma^k v_{t+k} \tag{3}$$

, where $r_{t+i}$ is the reward predicted from unrolled state $\hat{s}_{t+i}$. We name the sum of rewards $\sum_{i=0}^{k-1} \gamma^i r_{t+i}$ as the value prefix, since it is used as a prefix in the later Q-value computation.

We propose to predict value prefix from the unrolled states $(s_t, \hat{s}_{t+1}, \cdots, \hat{s}_{t+k-1})$ in an end-to-end manner, i.e. value-prefix $= f(s_t, \hat{s}_{t+1}, \cdots, \hat{s}_{t+k-1})$. Here $f$ is some neural network architecture that takes in a variable number of inputs and outputs a scalar. We choose the LSTM in our experiment. During the training time, the LSTM is supervised at every time step, since the value prefix can be computed whenever a new state comes in. This per-step rich supervision allows the LSTM can be trained well even with limited data. Compared with the naive per step reward prediction and summation approach, the end-to-end value prefix prediction is more accurate, because it can automatically handle the intermediate state aliasing problem. See Experiment Section 5.3 for empirical evaluations. As a result, it helps the MCTS to explore better, and thus increases the performance. See the Appendix for architectural details.

### 4.3 Model-Based Off-Policy Correction

In MCTS RL algorithms, the value function fits the value of the current neural network policy. However, in practice as MuZero Reanalyze does, the value target is computed by sampling a trajectory from the replay buffer and computing: $z_t = \sum_{i=0}^{k-1} \gamma^i u_{t+i} + \gamma^k v_{t+k}$. This value target suffers from off-policy issues, since the trajectory is rolled out using an older policy, and thus the value target is no longer accurate. When data is limited, we have to reuse the data sampled from a much older policy, thus exaggerating the inaccurate value target issue.

In previous model-free settings, there is no straightforward approach to fix this issue. On the contrary, since we have a model of the environment, we can use the model to imagine an "online experience". More specifically, we propose to use rewards of a dynamic horizon $l$ from the old trajectory, where $l < k$ and $l$ should be smaller if the trajectory is older. This reduces the policy divergence by fewer rollout steps. Further, we redo an MCTS search with the current policy on the last state $s_{t+l}$ and compute the empirical mean value at the root node. This effectively corrects the off policy issue using imagined rollouts with current policy and reduces the increased bias caused by setting $l$ less than $k$. Formally, we propose to use the following value target:

$$z_t = \sum_{i=0}^{l-1} \gamma^i u_{t+i} + \gamma^l \nu_{t+l}^{\text{MCTS}} \tag{4}$$

where $l <= k$ and the older the sampled trajectory, the smaller the $l$. $\nu^{\text{MCTS}}(s_{t+l})$ is the root value of the MCTS tree expanded from $s_{t+l}$ with the current policy, as MuZero non-Reanalyze does. See the Appendix for how to choose $l$. In practice, the computation cost of the correction is two times on the reanalyzed side. However, the training will not be affected due to the parallel implementation.

# 5 Experiments

In this section, we aim to evaluate the sample efficiency of the proposed algorithm. Here, the sample efficiency is measured by the performance of each algorithm at a common, small amount of environment transitions, i.e. the better the performance, the higher the sample efficiency. More specifically, we use the Atari 100k benchmark. Intuitively, this benchmark asks the agent to learn to play Atari games within two hours of real-world game time. Additionally, we conduct some ablation studies to investigate and analyze each component on Atari 100k. To further show the sample efficiency, we apply EfficientZero to some simulated robotics environments on the DMControl 100k benchmark, which contains the same 100k environment steps.

## 5.1 Environments

**Atari 100k** Atari 100k was first proposed by the SimPLe [18] method, and is now used by many sample-efficient RL works, such as Srinivas et al. [35], Laskin et al. [22], Kostrikov et al. [21], Schwarzer et al. [32]. The benchmark contains 26 Atari games, and the diverse set of games can effectively measure the performance of different algorithms. The benchmark allows the agent to interact with 100 thousand environment steps, i.e. 400 thousand frames due to a frameskip of 4, with each environment. 100k steps roughly correspond to 2 hours of real-time gameplay, which is far less than the usual RL settings. For example, DQN [24] uses 200 million frames, which is around 925 hours of real-time gameplay. Note that the human player's performance is tested after allowing the human to get familiar with the game after 2 hours as well. We report the raw performance on each game, as well as the mean and median of the human normalized score. The human normalized score is defined as: $(\text{score}_{\text{agent}} - \text{score}_{\text{random}})/(\text{score}_{\text{human}} - \text{score}_{\text{random}})$.

We compare our method to the following baselines. (1) SimPLe [18], a model-based RL algorithm that learns an action conditional video prediction model and trains PPO within the learned environment. (2) OTRainbow [19], which tunes the hyper-parameters of the Rainbow [16] method to achieve higher sample efficiency. (3) CURL [35], which uses contrastive learning as a side task to improve the image representation quality. (4) DrQ [21], which adds data augmentations to the input images while learning the original RL objective. (5) SPR [32], the previous SoTA in Atari 100k which proposes to augment the Rainbow [16] agent with data augmentations as well as a multi-step consistency loss using BYOL-style self-supervision. (6) MuZero [27] with our implementations and the same hyper-parameters as EfficientZero. (7) Random Agent (8) Human performance.

**DeepMind Control 100k** Tassa et al. [37] propose the DMControl suite, which includes some challenging visual robotics tasks with continuous action space. And some works [12, 35] have benchmarked for the sample efficiency on the DMControl 100k which contains 100k environment steps data. Since the MCTS-based methods cannot deal with tasks with continuous action space, we discretize each dimension into 5 discrete slots in MuZero [27] and EfficientZero. To avoid the dimension explosion, we evaluate EfficientZero in three low-dimensional tasks.

We compare our method to the following baselines. (1) Pixel SAC, which applies SAC directly to pixels. (2) SAC-AE [42], which combines the SAC and an auto-encoder to handle image-based inputs. (3) State SAC, which applies SAC directly to ground truth low dimensional states rather than the pixels. (4) Dreamer [12], which learns a world model and is trained in dreamed scenarios. (5) CURL [35], the previous SoTA in DMControl 100k. (6) MuZero [27] with action discretizations.

## 5.2 Results

Table 1 shows the results of EfficientZero on the Atari 100k benchmark. Normalizing our score with the score of human players, EfficientZero achieves a mean score of 1.904 and a median score of 1.160. As a reference, DQN [24] achieves a mean and median performance of 2.20 and 0.959 on these 26 games. However, it is trained with 500 times more data (200 million frames). For the first time, an agent trained with only 2 hours of game data can outperform the human player in terms of the mean and median performance. Among all games, our method outperforms the human in 14 out of 26 games. Compared with the previous state-of-the-art method (SPR [32]), we are 170% and 180% better in terms of mean and median score respectively.

Apart from the Atari games, EffcientZero achieves remarkable results in the simulated tasks with continuous action space. As shown in Table 2, EffcientZero outperforms CURL, the previous SoTA,

to a considerable degree and keeps a smaller variance but MuZero cannot work well here. Notably, EfficientZero achieves comparable results to the state SAC, which consumes the ground truth states as input and is considered as the oracles.

Table 1: Scores achieved on the Atari 100k benchmark (32 seeds). EfficientZero achieves super-human performance with only 2 hours of real-time game play. Our method is 170% and 180% better than the previous SoTA performance, in mean and median human normalized score respectively.

| Game | Random | Human | SimPLe | OTRainbow | CURL | DrQ | SPR | MuZero | Ours |
|---|---|---|---|---|---|---|---|---|---|
| Alien | 227.8 | 7127.7 | 616.9 | 824.7 | 558.2 | 771.2 | 801.5 | 530.0 | **1140.3** |
| Amidar | 5.8 | 1719.5 | 88.0 | 82.8 | 142.1 | 102.8 | **176.3** | 38.8 | 101.9 |
| Assault | 222.4 | 742.0 | 527.2 | 351.9 | 600.6 | 452.4 | 571.0 | 500.1 | **1407.3** |
| Asterix | 210.0 | 8503.3 | 1128.3 | 628.5 | 734.5 | 603.5 | 977.8 | 1734.0 | **16843.8** |
| Bank Heist | 14.2 | 753.1 | 34.2 | 182.1 | 131.6 | 168.9 | **380.9** | 192.5 | 361.9 |
| BattleZone | 2360.0 | 37187.5 | 5184.4 | 4060.6 | 14870.0 | 12954.0 | 16651.0 | 7687.5 | **17938.0** |
| Boxing | 0.1 | 12.1 | 9.1 | 2.5 | 1.2 | 6.0 | 35.8 | 15.1 | **44.1** |
| Breakout | 1.7 | 30.5 | 16.4 | 9.8 | 4.9 | 16.1 | 17.1 | 48.0 | **406.5** |
| ChopperCmd | 811.0 | 7387.8 | 1246.9 | 1033.3 | 1058.5 | 780.3 | 974.8 | 1350.0 | **1794.0** |
| Crazy Climber | 10780.5 | 35829.4 | 62583.6 | 21327.8 | 12146.5 | 20516.5 | 42923.6 | 56937.0 | **80125.3** |
| Demon Attack | 152.1 | 1971.0 | 208.1 | 711.8 | 817.6 | 1113.4 | 545.2 | 3527.0 | **13298.0** |
| Freeway | 0.0 | 29.6 | 20.3 | 25.0 | **26.7** | 9.8 | 24.4 | 21.8 | 21.8 |
| Frostbite | 65.2 | 4334.7 | 254.7 | 231.6 | 1181.3 | 331.1 | **1821.5** | 255.0 | 313.8 |
| Gopher | 257.6 | 2412.5 | 771.0 | 778.0 | 669.3 | 636.3 | 715.2 | 1256.0 | **3518.5** |
| Hero | 1027.0 | 30826.4 | 2656.6 | 6458.8 | 6279.3 | 3736.3 | 7019.2 | 3095.0 | **8530.1** |
| Jamesbond | 29.0 | 302.8 | 125.3 | 112.3 | **471.0** | 236.0 | 365.4 | 87.5 | 459.4 |
| Kangaroo | 52.0 | 3035.0 | 323.1 | 605.4 | 872.5 | 940.6 | **3276.4** | 62.5 | 962.0 |
| Krull | 1598.0 | 2665.5 | 4539.9 | 3277.9 | 4229.6 | 4018.1 | 3688.9 | 4890.8 | **6047.0** |
| Kung Fu Master | 258.5 | 22736.3 | 17257.2 | 5722.2 | 14307.8 | 9111.0 | 13192.7 | 18813.0 | **31112.5** |
| Ms Pacman | 307.3 | 6951.6 | **1480.0** | 941.9 | 1465.5 | 960.5 | 1313.2 | 1265.6 | 1387.0 |
| Pong | -20.7 | 14.6 | 12.8 | 1.3 | -16.5 | -8.5 | -5.9 | -6.7 | **20.6** |
| Private Eye | 24.9 | 69571.3 | 58.3 | 100.0 | **218.4** | -13.6 | 124.0 | 56.3 | 100.0 |
| Qbert | 163.9 | 13455.0 | 1288.8 | 509.3 | 1042.4 | 854.4 | 669.1 | 3952.0 | **15458.1** |
| Road Runner | 11.5 | 7845.0 | 5640.6 | 2696.7 | 5661.0 | 8895.1 | 14220.5 | 2500.0 | **18512.5** |
| Seaquest | 68.4 | 42054.7 | 683.3 | 286.9 | 384.5 | 301.2 | 583.1 | 208.0 | **1020.5** |
| Up N Down | 533.4 | 11693.2 | 3350.3 | 2847.6 | 2955.2 | 3180.8 | **28138.5** | 2896.9 | 16095.7 |
| Normed Mean | 0.000 | 1.000 | 0.443 | 0.264 | 0.381 | 0.357 | 0.704 | 0.562 | **1.904** |
| Normed Median | 0.000 | 1.000 | 0.144 | 0.204 | 0.175 | 0.268 | 0.415 | 0.227 | **1.160** |

Table 2: Scores achieved by EfficientZero (mean & standard deviation for 10 seeds) and some baselines on some low-dimensional environments on the DMControl 100k benchmark. EfficientZero achieves state-of-art performance and comparable results to the state-based SAC.

| Task | CURL | Dreamer | MuZero | SAC-AE | Pixel SAC | State SAC | EfficientZero |
|---|---|---|---|---|---|---|---|
| Cartpole, Swingup | 582± 146 | 326±27 | 218.5 ± 122 | 311±11 | 419±40 | 835±22 | **813±19** |
| Reacher, Easy | 538± 233 | 314±155 | 493 ± 145 | 274±14 | 145±30 | 746±25 | **952±34** |
| Ball in cup, Catch | 769± 43 | 246 ± 174 | 542 ± 270 | 391± 82 | 312± 63 | 746±91 | **942±17** |

## 5.3 Ablations

In Section 4, we discuss three issues that prevent MuZero from achieving high performance when data is limited: (1) the lack of environment model supervision, (2) the state aliasing issue, and (3) the off-policy target value issue. We propose three corresponding approaches to fix those issues and demonstrate the usefulness of the combination of those approaches on a wide range of 26 Atari games. In this section, we will analyze each component individually.

**Each Component** Firstly, we do an ablation study by removing the three components from our full model one at a time. As shown in Table 3, we find that removing any one of the three components will lead to a performance drop compared to our full model. Furthermore, the richer learning signals are the aspect Muzero lacks most in the low-data regime as the largest performance drop is from the version without consistency supervision. As for the performance in the high-data regime, We find that the temporal consistency can significantly accelerate the training. The value prefix seems to be helpful during the early learning process, but not as much in the later stage. The off-policy correction is not necessary as it is specifically designed under limited data.

Table 3: Ablations of the self-supervised consistency, end-to-end value prefix and model-based off-policy correction. We remove one component at a time and evaluate the corresponding version on the 26 Atari games. Each component matters and the consistency one is the most significant. The detailed results are attached in the Appendix .

| Game | Full | w.o. consistency | w.o. value prefix | w.o. off-policy correction |
|---|---|---|---|---|
| Normed Mean | **1.904** | 0.881 | 1.482 | 1.475 |
| Normed Median | **1.160** | 0.340 | 0.552 | 0.836 |

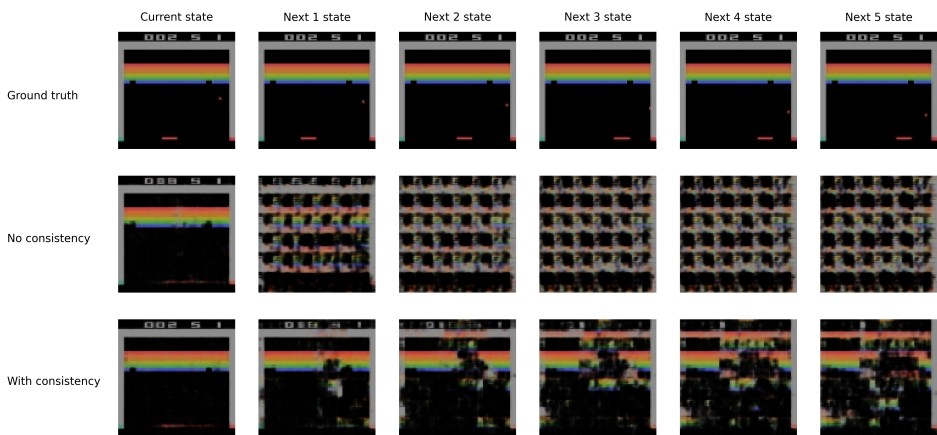

Figure 4: Evaluations of image reconstructions based on latent states extracted from the model with or without self-supervised consistency. The predicted next states with consistency can basically be reconstructed into observations while the ones without consistency cannot.

**Temporal Consistency** As the version without self-supervised consistency cannot work well in most of the games, we attempt to dig into the reason for such phenomenon. We design a decoder $\mathcal{D}$ to reconstruct the original observations, taking the latent states as inputs. Specifically, the architecture of $\mathcal{D}$ and the $\mathcal{H}$ are symmetrical, which means that all the convolutional layers are replaced by deconvolutional layers in $\mathcal{D}$ and the order of the layers are reversed in $\mathcal{D}$. Therefore, $\mathcal{H}$ is an encoder to obtain state $s_t$ from observation $o_t$ and $\mathcal{D}$ tries to decode the $o_t$ from $s_t$. In this ablation, we freeze all parameters of the trained EfficientZero network with or without consistency respectively and the reconstructed results are shown in different columns of Figure 4. We regard the decoder as a tool to visualize the current states and unrolled states, shown in different rows of Figure 4. Here we note that $\mathcal{M}_{\text{con}}$ is the trained EfficientZero model with consistency and $\mathcal{M}_{\text{non}}$ is the one without consistency.

As shown in Figure 4, in terms of the current state $s_t$, the observation is reconstructed well enough in the two versions. However, it is remarkable that the the decoder given $\mathcal{M}_{\text{non}}$ can not reconstruct images from the unrolled predicted states $\hat{s}_{t+k}$ while the one given $\mathcal{M}_{\text{con}}$ can reconstruct the basic observations.

To sum up, there are some distributional shifts between the latent states from the representation network and the states from the dynamics function without consistency. The consistency component can reduce the shift and provide more supervision for training the dynamics network.

**Value Prefix** We further validate our assumptions in the end-to-end learning of value prefix, i.e. the state aliasing problem will cause difficulty in predicting the reward, and end-to-end learning of value prefix can alleviate this phenomenon.

To fairly compare directly predicting the reward versus end-to-end learning of the value prefix, we need to control for the dataset that both methods are trained on. Since during the RL training, the dataset distribution is determined by the method, we opt to load a half-trained Pong model and rollout total 100k steps as the common static dataset. We split this dataset into a training set and a validation set. Then we run both the direct reward prediction and the value prefix method on the training split.

As shown in Figure 5, we find that the direct reward prediction method has lower losses on the training set. However, the value prefix's validation error is much smaller when unrolled for 5 steps. This shows that the value prefix method avoids overfitting the hard reward prediction problem, and thus it can reduce the state aliasing problem, reaching a better generalization performance.

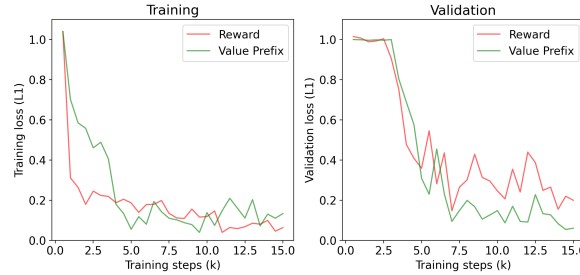

Figure 5: Training and validation losses of direct reward prediction method and the value prefix method.

**Off-Policy Correction** To prove the effectiveness of the off-policy correction component, we compare the error between the target values and the ground truth values with or without off-policy correction. Specifically, the ground truth values are estimated by Monte Carlo sampling.

We train a model for the game UpNDown with total 100k training steps, and collect the trajectories at different training stages respectively (20k, 40k, ..., 100k steps). Then we calculate the ground truth values with the final model. We choose the trajectories at the same stage (20k) and use the final model to evaluate the target values with or without off-policy correction, following the Equation 4. We evaluate the L1 error of the target values and the ground truth, as shown in Table 4. The error of unrolled next 5 states means the average error of the unrolled 1-5 states with dynamics network from current states. The error is smaller in both current states and the unrolled states with off-policy correction. Thus, the correction component does reduce the bias caused by the off-policy issue.

Table 4: Ablations of the off-policy correction: L1 error of the target values versus the ground truth values. Take UpNDown as an example.

| States | Current state | Unrolled next 5 states (Avg.) | All states (Avg.) |
|---|---|---|---|
| Value error without correction | 0.765 | 0.636 | 0.657 |
| Value error with correction | **0.533** | **0.576** | **0.569** |

Furthermore, we also ablate the value error of the trajectories at distinct stages in Table 5. We can find that the value error becomes smaller as the trajectories are fresher. This indicates that the off-policy issue is severe due to the staleness of the data. More significantly, the off-policy correction can provide more accurate target value estimation for the trajectories at distinct time-steps as all the errors with correction shown in the table are smaller than those without correction at the same stage.

Table 5: Ablations of the off-policy correction: Average L1 error of the values of the trajectories at distinct stages. Take UpNDown as an example.

| Stages of trajectories | 20k | 40k | 60k | 80k | 100k |
|---|---|---|---|---|---|
| Value error without correction | 0.657 | 0.697 | 0.628 | 0.574 | 0.441 |
| Value error with correction | **0.569** | **0.552** | **0.537** | **0.488** | **0.397** |

# 6    Discussion

In this paper, we propose a sample-efficient model-based method EfficientZero. It achieves super-human performance on the Atari games with as little as 2 hours of the gameplay experience and state-of-the-art performance on some DMControl tasks. Apart from the full results, we do detailed ablation studies to examine the effectiveness of the proposed components. This work is one step towards running RL in the physical world with complex sensory inputs. In the future, we plan to extend it to more directions, such as a better design for the continuous action space. And we also plan to study the acceleration of MCTS and how to combine this framework with life-long learning.

## Acknowledgments and Disclosure of Funding

This work is supported by the Ministry of Science and Technology of the People's Republic of China, the 2030 Innovation Megaprojects "Program on New Generation Artificial Intelligence" (Grant No. 2021AAA0150000).

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
