# Mastering Atari Games with Limited Data

**Weirui Ye**[*]   **Shaohuai Liu**[*]   **Thanard Kurutach**[†]   **Pieter Abbeel**[†]   **Yang Gao**[*‡]
[*]Tsinghua University, [†]UC Berkeley, [‡] Shanghai Qi Zhi Institute

## A   Appendix

### A.1   Models and Hyper-parameters

As for the architecture of the networks, there are three parts in our model pipeline: the representation part, the dynamics part, and the prediction part. The architecture of the representation part is as follows:

- 1 convolution with stride 2 and 32 output planes, output resolution 48x48. (BN + ReLU)
- 1 residual block with 32 planes.
- 1 residual downsample block with stride 2 and 64 output planes, output resolution 24x24.
- 1 residual block with 64 planes.
- Average pooling with stride 2, output resolution 12x12. (BN + ReLU)
- 1 residual block with 64 planes.
- Average pooling with stride 2, output resolution 6x6. (BN + ReLU)
- 1 residual block with 64 planes.

, where the kernel size is $3 \times 3$ for all operations.

As for the dynamics network, we follow the architecture of MuZero [7] but reduce the residual blocks from 16 to 1. Furthermore, we add an extra residual link in the dynamics part to keep the information of historical hidden states during recurrent inference. The design of the dynamics network is listed here:

- Concatenate the input states and input actions into 65 planes.
- 1 convolution with stride 2 and 64 output planes. (BN)
- A residual link: add up the output and the input states. (ReLU)
- 1 residual block with 64 planes.

In the prediction part, we use two-layer MLPs with batch normalization to predict the reward, value, or policy. Considering the stability of the prediction part, we set the weights and bias of the last layer to zero in prediction networks. As for the reward prediction network, it predicts the sum of the rewards, namely value prefix: $r_t, h_{t+1} = \mathcal{R}(\hat{s}_{t+1}, h_t)$, where $r_t$ is the predicted sum of rewards, $h_0$ is zero-initialized and hidden size of LSTM is 512. The architecture of the value prediction network is as follows:

- 1 1x1convolution and 16 output planes. (BN + ReLU)
- Flatten.
- LSTM with 512 hidden size. (BN + ReLU)

---

[*]{ywr20, liush20}@mails.tsinghua.edu.cn, gaoyangiiis@tsinghua.edu.cn
[†]{thanard.kurutach, pabbeel}@berkeley.edu

35th Conference on Neural Information Processing Systems (NeurIPS 2021), Sydney, Australia.

- 1 fully connected layers and 32 output dimensions. (BN + ReLU)
- 1 fully connected layers and 601 output dimensions.

The horizontal length of the LSTM during training is limited to the unrolled steps $l_{\text{unroll}} = 5$, but it will be larger in MCTS as the dynamics process can go deeper. Therefore, we reset the hidden state of LSTM after $\zeta = 5$ steps of recurrent inference, where $\zeta$ is the valid horizontal length.

The design of the reward and policy prediction networks are the same except for the dimension of the outputs:

- 1 residual block with 64 planes.
- 1 1x1convolution and 16 output planes. (BN + ReLU)
- Flatten.
- 1 fully connected layers and 32 output dimensions. (BN + ReLU)
- 1 fully connected layers and $D$ output dimensions.

, where $D = 601$ in the reward prediction network and $D$ is equal to the action space in the policy prediction network.

Here is the brief introduction of the training pipeline, taking one-step rollout as an example.

$$
\begin{aligned}
s_t &= \mathcal{H}(o_t) \\
s_{t+1} &= \mathcal{H}(o_{t+1}) \\
\hat{s}_{t+1} &= \mathcal{G}(s_t, a_t) \\
v_t &= \mathcal{V}(s_t) \\
p_t &= \mathcal{P}(s_t) \\
r_t, h_{t+1} &= \mathcal{R}(\hat{s}_{t+1}, h_t) = \mathcal{R}(\mathcal{G}(s_t, a_t), h_t)
\end{aligned}
\tag{1}
$$

, where $\mathcal{H}$ is the representation network, $\mathcal{G}$ is the dynamics network, $\mathcal{V}$ is the value prediction network, $\mathcal{P}$ is the policy prediction network, $\mathcal{R}$ is the reward (*value prefix*) prediction network. $o_t, s_t, a_t$ are observations, states and actions. $h_t$ is the hidden states in recurrent neural networks.

Here is the training loss, taking one-step rollout as an example:

$$
\begin{aligned}
\mathcal{L}_{\text{similarity}}(s_{t+1}, \hat{s}_{t+1}) &= \mathcal{L}_2(sg(P_1(s_{t+1})), P_2(P_1(\hat{s}_{t+1}))) \\
\mathcal{L}_t(\theta) &= \mathcal{L}(u_t, r_t) + \lambda_1 \mathcal{L}(\pi_t, p_t) + \lambda_2 \mathcal{L}(z_t, v_t) \\
&\quad + \lambda_3 \mathcal{L}_{\text{similarity}}(s_{t+1}, \hat{s}_{t+1}) + c||\theta||^2 \\
\mathcal{L}(\theta) &= \frac{1}{l_{\text{unroll}}} \sum_{i=0}^{l_{\text{unroll}}-1} \mathcal{L}_{t+i}(\theta)
\end{aligned}
\tag{2}
$$

, where $\mathcal{L}$ is the total loss of the unrolled $l_{\text{unroll}}$ steps, $\mathcal{L}_1$ is the Cross-Entropy loss, and $\mathcal{L}_2$ is the negtive cosine similarity loss. Besides, $P_1$ is a 3-layer MLP while $P_2$ is a 2-layer MLP. The dimension of the hidden layers is 512 and the dimension of the output layers is 1024. We add batch normalization between every two layers in those MLP except the final layer. $sg(P_1)$ means stopping gradients.

We stack 4 historical frames, with an interval of 4 frames-skip. Thus the input effectively covers 16 frames of the game history. We stack the input images on the channel dimension, resulting in a $96 \times 96 \times 12$ tensor. We do not use any extra state normalization besides the batch norm and we choose reward clipping to keep better scales in the searching process.

Generally, compared with MuZero [7], we reduce the number of residual blocks and the number of planes as we find that there is no capability issue caused by much smaller networks in our EfficientZero with limited data. In another word, such a tiny network can acquire good performance in the limited setting.

For other details, we provide hyper-parameters in Table 1. It is notable that we train the model for 120k steps where we only collect data during the first 100k steps. In this way, the latter trajectories can be fully used in training. Besides, the learning rate will drop after every 100k training steps (from 0.2 to 0.02 at 100k).

Table 1: Hyper-parameters for EfficientZero on Atari games

| Parameter | Setting |
|---|---|
| Observation down-sampling | $96 \times 96$ |
| Frames stacked | 4 |
| Frames skip | 4 |
| Reward clipping | True |
| Terminal on loss of life | True |
| Max frames per episode | 108K |
| Discount factor | $0.997^4$ |
| Minibatch size | 256 |
| Optimizer | SGD |
| Optimizer: learning rate | 0.2 |
| Optimizer: momentum | 0.9 |
| Optimizer: weight decay ($c$) | 0.0001 |
| Learning rate schedule | $0.2 \to 0.02$ |
| Max gradient norm | 5 |
| Priority exponent ($\alpha$) | 0.6 |
| Priority correction ($\beta$) | $0.4 \to 1$ |
| Training steps | 120K |
| Evaluation episodes | 32 |
| Min replay size for sampling | 2000 |
| Self-play network updating inerval | 100 |
| Target network updating interval | 200 |
| Unroll steps ($l_{unroll}$) | 5 |
| TD steps ($k$) | 5 |
| Policy loss coefficient ($\lambda_1$) | 1 |
| Value loss coefficient ($\lambda_2$) | 0.25 |
| Self-supervised consistency loss coefficient ($\lambda_3$) | 2 |
| LSTM horizontal length ($\zeta$) | 5 |
| Dirichlet noise ratio ($\xi$) | 0.3 |
| Number of simulations in MCTS ($N_{sim}$) | 50 |
| Reanalyzed policy ratio | 0.99 |

### A.2 More Ablations

In the experiment section , we list some ablation studies to prove the effectiveness of each component. In this section, we will display more results for the ablation study.

Firstly, the detailed results of the ablation study of each component are listed in Table 2. In this table, We find that the full version of EfficientZero outperforms the others without any one of the components. Furthermore, for those environments EfficientZero can already solve, the performance is similar between the full version and the version without off-policy correction, such as Breakout, Pong, etc. In such a case, the off-policy issue is not severe, which is the reason for this phenomenon. Besides, for some environments with sparse rewards, the value prefix component matters, such as Pong; and for those with dense rewards, the state aliasing problem has less negative effects for the reward signals are sufficient, such as Qbert. As for the version without self-supervised consistency, the results of all the environments are much poorer.

In addition, we do the ablation study for the data augmentation technique in the consistency component to examine the effect of data augmentations. We apply a random small shift of 0-4 pixels as well as the change of the intensity as the augmentation techniques. Here we choose several Atari games and train the model for 100k steps. The results are shown in Table 3. We can find that the version without data augmentation has similar performances while the version without consistency component is worse. This indicates that the improvement of the consistency component is basically from the self-supervised learning loss rather than the data augmentation.

Finally, we also do the ablation study for the MCTS root value and the dynamic horizon in the off-policy correction component. Here we choose several Atari games and train the model for 100k steps. As shown in Table 4, the version without dynamic horizon has poorer results than that without

Table 2: Ablations of the self-supervised consistency, end-to-end value prefix and model-based off-policy correction on more Atari games. (Scores on the Atari 100k benchmark)

| Game | Full | w.o. consistency | w.o. value prefix | w.o. off-policy correction |
|---|---|---|---|---|
| Alien | 808.5 | **961.3** | 558 | 619.4 |
| Amidar | 148.6 | 32.2 | 31.0 | **256.3** |
| Assault | **1263.1** | 572.9 | 955.0 | 1190.4 |
| Asterix | **25557.8** | 2065.6 | 7330.0 | 13525.0 |
| Bank Heist | **351.0** | 165.6 | 273.0 | 297.5 |
| BattleZone | 13871.2 | 14063.0 | 9900.0 | **16125.0** |
| Boxing | 52.7 | 6.1 | **60.2** | 30.5 |
| Breakout | **414.1** | 237.4 | 379.2 | 400.3 |
| ChopperCommand | 1117.3 | 1138.0 | 1280 | **1487.5** |
| Crazy Climber | 83940.2 | 75550.0 | **106090.0** | 70681.0 |
| Demon Attack | **13003.9** | 5973.8 | 6818.5 | 8640.6 |
| Freeway | **21.8** | 21.8 | 21.8 | 21.8 |
| Frostbite | **296.3** | 248.8 | 235.2 | 227.5 |
| Gopher | **3260.3** | 1155 | 2792.0 | 2275.0 |
| Hero | **9315.9** | 5824.4 | 3167.5 | 9053.0 |
| Jamesbond | **517.0** | 154.7 | 380.0 | 356.3 |
| Kangaroo | **724.1** | 375.0 | 200.0 | 687.5 |
| Krull | **5663.3** | 4178.625 | 4527.6 | 3635.6 |
| Kung Fu Master | **30944.8** | 19312.5 | 25980.0 | 25025.0 |
| Ms Pacman | 1281.2 | 1090.0 | **1475.0** | 1297.2 |
| Pong | **20.1** | -1.5 | 16.8 | 19.5 |
| Private Eye | 96.7 | **100.0** | 100.0 | 100.0 |
| Qbert | **13781.9** | 5340.7 | 6360.0 | 13637.5 |
| Road Runner | **17751.3** | 2700.0 | 3010.0 | 9856.0 |
| Seaquest | **1100.2** | 460.0 | 468.0 | 843.8 |
| Up N Down | **17264.2** | 3040.0 | 7656.0 | 4897.2 |
| Normed Mean | **1.943** | 0.881 | 1.482 | 1.475 |
| Normed Median | **1.090** | 0.340 | 0.552 | 0.836 |

Table 3: Ablations of the data augmentation technique in the consistency component. Results show that the data augmentation has limited improvement in EfficientZero and the self-supervised training loss is more significant.

| Game | Full | w.o. consistency | w.o. data augmentation |
|---|---|---|---|
| Asterix | 6218.8 | 1350.0 | **13884.0** |
| Breakout | **388.8** | 12.0 | 365.2 |
| Demon Attack | **10536.6** | 5973.8 | 8730.0 |
| Gopher | **2828.8** | 1155.0 | 1823.75 |
| Pong | **19.8** | -8.5 | 13.9 |
| Qbert | **15268.8** | 2304.7 | 14286.0 |
| Seaquest | **1321.0** | 460.0 | 1125.0 |
| Up N Down | 10238.1 | 3040.0 | **16380.0** |

the MCTS root value. In the off-policy correction component, the dynamic horizon seems more important.

### A.3 MCTS Details

Our policy searching approach is based on Monte-Carlo tree search (MCTS). We follow the procedure in MuZero [7], which includes three stages and repeats the searching process for $N_{\text{sim}} = 50$ simulations. Here are some brief introductions for each stage.

**Selection** In the selection part, it targets at choosing an appropriate unvisited node while balancing exploration and exploitation with UCT:

$$
a^k = \left\{ \arg\max_a Q(s,a) + P(s,a) \frac{\sqrt{\sum_b N(s,b)}}{1 + N(s,a)} \left( c_1 + \log\left( \frac{\sum_b N(s,b) + c_2 + 1}{c_2} \right) \right) \right\} \quad (3)
$$

, where $Q(s,a)$ is the average Q values after simulations, $N(s,a)$ is the total visit counts at state $s$ by selecting action $a$, and $P(s,a)$ is the policy prior set in the expansion process. In each simulation, the

Table 4: Ablations of the techniques (the MCTS root value and the dynamic horizon) in the off-policy correction component. The dynamic horizon seems more important than the MCTS root value when data is limited.

| Game | Full | w.o. off-policy correction | w.o. dynamic horizon | w.o. MCTS root value |
|------|------|------|------|------|
| Asterix | 6218.8 | 2706.3 | 3263.0 | **6288.0** |
| Breakout | 388.8 | **468.6** | 427.0 | 387.8 |
| Demon Attack | **10536.6** | 8640.6 | 9211.1 | 10063.0 |
| Gopher | **2828.8** | 2275.0 | 2459.2 | 2651.0 |
| Pong | **19.8** | 19.5 | 19.2 | 14.5 |
| Qbert | **15268.8** | 3948.4 | 7945 | 14738.0 |
| Seaquest | **1321.0** | 1248.0 | 1292.0 | 876.0 |
| Up N Down | **10238.1** | 3240.0 | 4772.0 | 9925.6 |

MCTS starts from the root node $s^0$. And for each time-step $k = 1...l$ of the simulation, the algorithm will select the action $a^k$ according to the UCT. Usually, $c_1 = 1.25$ and $c_2 = 19652$ according to the literature [8, 9, 3].

However, the default Q value of the unvisted node is set to 0, which indicates the worst state. To give a better Q-value estimation of the unvisited nodes, we evaluate a mean Q value mechanism in each simulation for tree nodes, similar to the implementation of Elf OpenGo [10].

$$\hat{Q}(s^{\text{root}}) = 0$$

$$\hat{Q}(s) = \frac{\hat{Q}(s^{\text{parent}}) + \sum_b \mathbf{1}_{N(s,b)>0} Q(s,b)}{1 + \sum_b \mathbf{1}_{N(s,b)>0}}$$

$$Q(s,a) := \begin{cases} Q(s,a) & N(s,a) > 0 \\ \hat{Q}(s) & N(s,a) = 0 \end{cases}$$

(4)

, where $\hat{Q}(s)$ is the estimated Q value for unvisited nodes to make better selections considering exploration and exploitation. $s^{\text{root}}$ is the state of the root node and $s^{\text{parent}}$ is the state of the parent node of $s$. In experiments, we find that the mean Q value mechanism gives a better exploration than the default one.

**Expansion** Then the newly selected node will be expanded with the predicted reward and policy as its prior. Furthermore, when the root node is to expand, we apply the Dirichlet noise to the policy prior during the self-play stage and the reanalyzing stage to give more explorations.

$$P(s,a) := (1 - \rho)P(s,a) + \rho \mathcal{N}_{\mathcal{D}}(\xi)$$

(5)

, where $\mathcal{N}_{\mathcal{D}}(\xi)$ is the Dirichlet noise distribution, $\rho, \xi$ is set to 0.25 and 0.3 respectively. However, we do not use any noise and set $\rho$ to 0 instead for those non-root node or during evaluations.

**Backup** After selecting and expanding a new node, we need to backup along the current searching trajectory to update the $Q(s,a)$. Considering the scales of values in distinct environments, we compute a normalized Q-value by using the minimum-maximum values calculated along with all visited tree nodes, which is applied in MuZero[7]. However, when the data is limited, the small difference between the minimum and maximum values will result in overconfidence in UCT calculation. For example, when all the Q-values in those visited tree nodes are in a range of 0 to $10^{-4}$, the normalized Q-value of $10^{-5}$ and $5 \times 10^{-5}$ will make a huge difference as one is normalized to 0.1 and another is 0.5. Therefore, we set a threshold here to reduce overconfidence in such occasions, which is called the soft minimum-maximum updates:

$$\bar{Q}(s^{k-1}, a^k) = \frac{Q(s^{k-1}, a^k) - \min_{(s,a)\in Tree} Q(s,a)}{\max(\max_{(s,a)\in Tree} Q(s,a) - \min_{(s,a)\in Tree} Q(s,a), \epsilon)}$$

(6)

, where $\epsilon$, the threshold to give a smooth range of the min-max bound, is set to 0.01.

After all the expansions in the MCTS, we will obtain average value and visit count distributions of the root node. Here, the root value can be applied in off-policy correction and the visit count distribution is the target policy distribution:

$$\pi(s,a) = \frac{N(s,a)^{1/T}}{\sum_b N(s,b)^{1/T}}$$

(7)

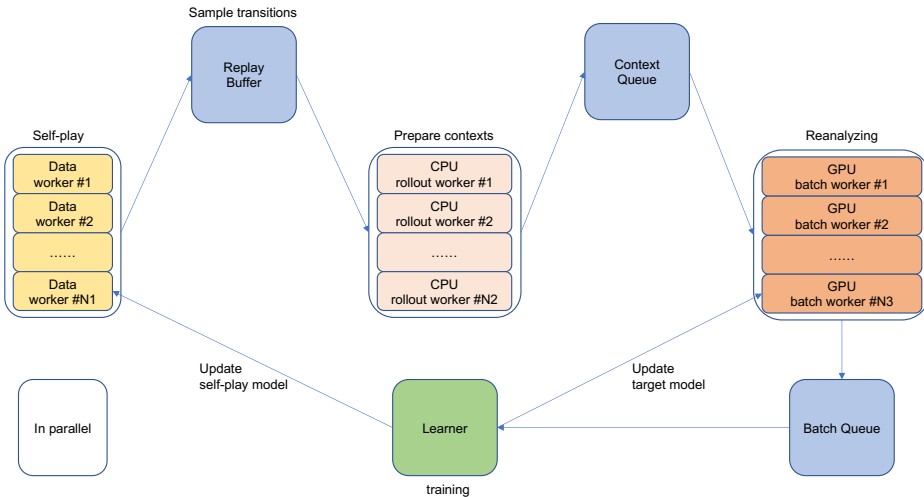

Figure 1: Pipeline of the EfficientZero implementation.

We decay the temperature of the MCTS output policy distribution here twice during training, at 50% and 75% of the training progress to 0.5 and 0.25 respectively.

## A.4 Training Details

In this subsection, we will introduce more training details.

**Pipeline** As for the code implementation of EfficientZero, we design a paralleled architecture with a double buffering mechanism in Pytorch and Ray, as shown in Figure 1.

Intuitively, we will describe the training process in a synchronized way. Firstly, the data workers called self-play actors are aimed at doing self-play with the given model updated within 600 training steps and then they will send the rolled-out trajectories into the replay buffer. Then the CPU rollout workers attempt to prepare the contexts of those batch transitions sampled from the replay buffer, in which way only CPU resources are required. Afterward, the GPU batch workers reanalyze those past data with the given contexts by the given target model, and most of the time-consuming parts in this procedure are in GPUs. Considering the frequent utilization of CPUs and GPUs in MCTS, the searching process is assigned for those GPU workers. Finally, the learner will obtain the reanalyzed batch and begin to train the agent.

The learner, all the data workers, CPU workers, and GPU workers start in parallel. The data workers and CPU workers share the replay buffer to sample data while the CPU and GPU workers share a context queue for reanalyzing data. Besides, the learner and the GPU workers use a batch queue to communicate. In such a design, we can utilize the CPU and GPU as much as possible.

**Self-play** During self-play, the priorities of the transition to collect are set to the max of the whole priorities in replay buffer. We also update the priority in EfficientZero according to MuZero [7]: $P(i) = \frac{p_i^\alpha}{\sum_k p_k^\alpha}$, where $p_i$ is the L1 error of the value during training. And the we scale with important sampling ratio $w_i = (\frac{1}{N \times P(i)})^\beta$. We set $\alpha$ to 0.6 and anneal $\beta$ from 0.4 to 1.0, following prioritized replay [6]. However, we find the priority mechanism only improves a little with limited data. Considering the long horizons in atari games, we collect the intermediate sequences of 400 moves.

**Reanalyze** The reanalyzed part is introduced in MuZero [7], which revisits the past trajectories and re-executes the data with lasted target model to obtain a fresher value and policy with model inference as well as MCTS.

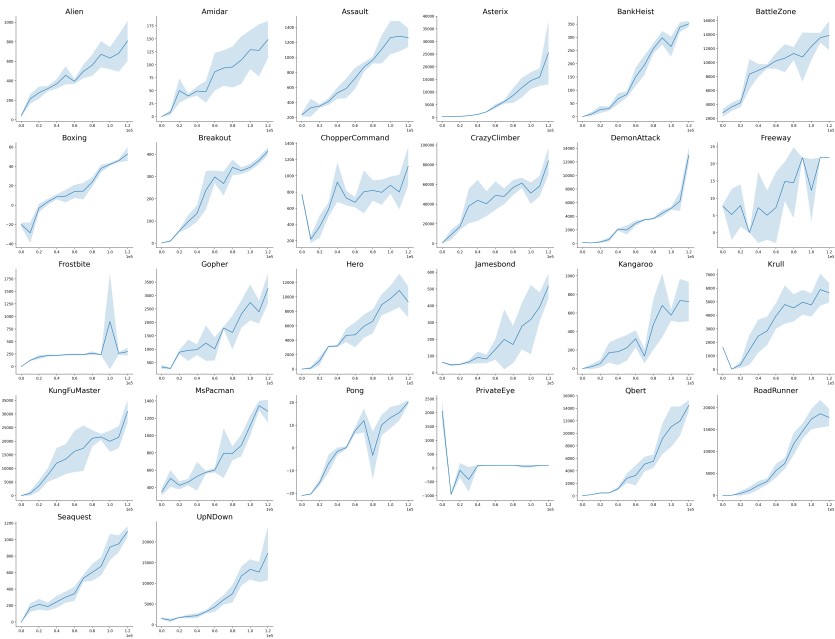

Figure 2: **Evaluation curves of *EfficientZero* on Atari 100k benchmark for individual games**. The average of the total rewards among 32 evaluation seeds for 3 runs is shown on the y-axis and the number of total training steps is 120,000, shown on the x-axis.

For the off-policy correction, the target values are reanalyzed as follows:

$$z_t = \sum_{i=0}^{l-1} \gamma^i u_{t+i} + \gamma^l \nu_{t+l}^{\text{MCTS}},$$

$$l = (k - \lfloor \frac{T_{\text{current}} - T_{s_t}}{\tau T_{\text{total}}} \rfloor).\text{clip}(1, k), l \in [1, k]$$

(8)

, where $k$ is the TD steps here, and is set to 5; $T_{\text{current}}$ is the current training steps, $T_{s_t}$ is the training steps of collecting the data $s_t$, $T_{\text{total}}$ is the total training steps (100k), and $\tau$ is a coefficient which is set to 0.3. Intuitively, $l$ is to define how fresh the collected data $s_t$ is. When the trajectory is stale, we need to unroll less to estimate the target values for the sake of the gaps between current model predictions and the stale trajectory rollouts. Besides, we replace the predicted value $v_{t+k}$ with the averaged root value from MCTS $\nu_{t+l}^{\text{MCTS}}$ to alleviate the off-policy bias.

Notably, we re-sample Dirichlet noise into the MCTS procedure in reanalyzed part to improve the sample efficiency with a more diverse searching process. Besides, we reanalyze the policy among 99% of the data and reanalyze the value among 100% data.

## A.5   Evaluation

We evaluate the EfficientZero on Atari 100k benchmark with a total of 26 games. Here are the evaluation curves during training, as shown in Figure 2.

Besides, we also report the scores for 3 runs (different seeds) with 32 evaluation seeds across the 26 Atari games, which is shown in Table 5.

Recently, Agarwal et al. [1] propose to use statistical tools to present more robust and efficient aggregate metrics. Here we display the corresponding results based on its open-sourced codebase. Figure 3 illustrates that EfficientZero significantly outperforms the other methods on Atari 100k benchmark concerning all the metrics.

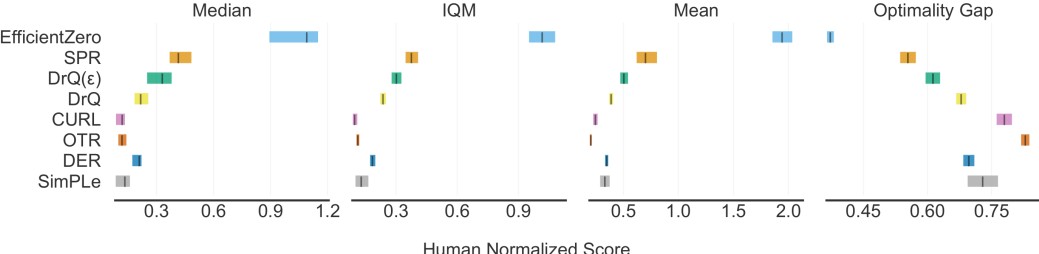

Figure 3: **Aggregate metrics on Atari 100k benchmark with 95% CIs**. Here the higher mean, median and IQM scores and lower optimality gap indicate better performance. The CIs are estimated by the percentile bootstrap with stratified sampling. All results except EfficientZero are from Agarwal et al. [1]. And all the methods are based on 10 runs per game except SimPLe with 5 runs and EfficientZero with 3 runs. EfficientZero significantly outperforms the other methods concerning the four metrics.

Table 5: Scores reported for 3 random seeds for each of the above games, with the last two columns being the mean and standard deviation across the runs. Each run is evaluated with 32 different seeds.

| Game | Seed 0 | Seed 1 | Seed 2 | Mean | Std |
|---|---|---|---|---|---|
| Alien | 1093.1 | 622.2 | 710.3 | 808.5 | 204.4 |
| Amidar | 198.7 | 116.4 | 130.6 | 148.6 | 35.9 |
| Assault | 1436.3 | 1150.8 | 1202.2 | 1263.1 | 124.3 |
| Asterix | 18421.9 | 43220.2 | 15031.3 | 25557.8 | 12565.7 |
| Bank Heist | 362.6 | 336.3 | 354.0 | 351.0 | 10.9 |
| Battle Zone | 11812.5 | 13100.8 | 16700.4 | 13871.2 | 2068.5 |
| Boxing | 45.9 | 49.9 | 62.4 | 52.7 | 7.0 |
| Breakout | 432.8 | 418.7 | 390.9 | 414.1 | 17.4 |
| ChopperCommand | 1190.9 | 1360.9 | 800.0 | 1117.3 | 234.8 |
| Crazy Climber | 98640.2 | 65520.4 | 87660.1 | 83940.2 | 13774.6 |
| Demon Attack | 11517.5 | 14323.3 | 13170.8 | 13003.9 | 1151.5 |
| Freeway | 21.8 | 21.8 | 21.8 | 21.8 | 0.0 |
| Frostbite | 407.1 | 225.5 | 256.3 | 296.3 | 79.4 |
| Gopher | 3002.6 | 2744.2 | 4034.1 | 3260.3 | 557.2 |
| Hero | 12349.1 | 8006.5 | 7592.0 | 9315.9 | 2151.5 |
| Jamesbond | 530.7 | 600.3 | 420.1 | 517.0 | 74.2 |
| Kangaroo | 980.2 | 460.7 | 731.3 | 724.1 | 212.1 |
| Krull | 4839.5 | 5548.5 | 6602.0 | 5663.3 | 724.1 |
| Kung Fu Master | 28493.1 | 36840.7 | 27500.5 | 30944.8 | 4188.7 |
| Ms Pacman | 1465.0 | 1203.4 | 1175.3 | 1281.2 | 130.4 |
| Pong | 20.6 | 18.8 | 21.0 | 20.1 | 1.0 |
| Private Eye | 100.0 | 90.0 | 100.0 | 96.7 | 4.7 |
| Qbert | 15458.1 | 14577.5 | 13310.0 | 14448.5 | 881.7 |
| Road Runner | 17843.8 | 20140.0 | 15270.2 | 17751.3 | 1989.2 |
| Seaquest | 1038.1 | 1078.2 | 1184.4 | 1100.2 | 61.7 |
| Up N Down | 22717.5 | 8095.6 | 20979.4 | 17264.2 | 6521.9 |

## A.6 Open Source EfficientZero Implementation

MCTS-based RL algorithms present a promising future research direction: to achieve strong performance with model-based methods. However, two major practical obstacles prevent them from being widely used currently. First, there are no high-quality open-source implementations of these algorithms. Existing implementations [11, 4] can only deal with simple state-based environments, such as CartPole [2]. Accurately scaling to complex image input environments requires non-trivial engineering efforts. Second, MCTS RL algorithms such as MuZero [7] require a large number of computations. For example, MuZero needs 64 TPUs to train 12 hours for one agent on Atari games. The high computational costs pose problems both for the future development of such methods as well as practical applications.

We think our open-source implementation of EfficientZero can drastically accelerate the research in MCTS RL algorithms. Our implementation is computationally friendly. To train an Atari agent

for 100k steps, it only needs 4 GPUs to train 7 hours. Our framework could potentially have a large impact on many real-world applications, such as robotics since it requires significantly fewer samples.

Our open-source framework aims to provide an easy way to understand the implementation while keeping relatively high compute efficiency. As shown in Fig. 4, the system is composed of four components: the replay buffer, the experience sampling actor, the reanalyze training target preparation module, and the training component.

To make sure the framework is easy to use, we implement them based on Ray [5], and the four components are implemented as ray actors which run in parallel. The main computation bottleneck is in the reanalyze module, which samples from the replay, and runs an MCTS search on each observation. To acceler-

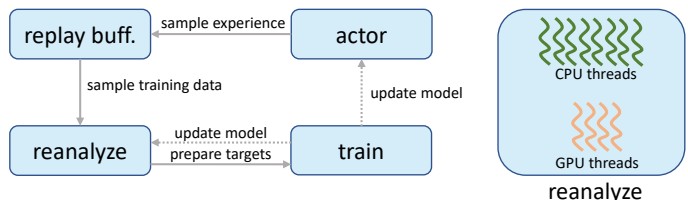

Figure 4: EfficientZero implementation overview.

ate the reanalyze module, we split the reanalyze computation into the CPU part and the GPU part, such that computation on CPU and GPU are run in parallel. We use a different number of actors between CPU and GPU to match their total throughput. To increase the throughput on GPU, we also collocate multiple batch computation threads on one GPU, as in Tian et al. [10]. We also implement the MCTS in C++ to avoid performance issues with Python on large amounts of atomic computations.

We implement the MCTS by a couple of important techniques, which are quite crucial to improve the efficiency of the MCTS process. On the one hand, we implement batch MCTS to allow the agent to search a batch of trees in parallel, to enlarge the throughput of MCTS during self-play and reanalyzing targets. On the other, we choose C++ in the MCTS process. However, the process of MCTS needs to do searching as well as model inference, which needs to communicate with Pytorch. Therefore, we use Python to do model inference, C++ to do other atomic computations, and Cython to communicate between Python contexts and C++ contexts. In another word, we use pure C++ to do selection, expansion, and backup while using neural networks in Python. Meanwhile, we build a database to store the hidden states in Python while storing the corresponding data index during the searching process in C++. For more details of the implementation, please refer to https://github.com/YeWR/EfficientZero.