# OpenReview forum: "Mastering Atari Games with Limited Data"
_NeurIPS.cc/2021/Conference — NeurIPS 2021 Poster_

### Official Review · Reviewer_CuAV · 2021-07-10

**Rating:** 7
**Confidence:** 4

**Summary:**

The authors propose EfficientZero, a sample efficient model-based visual RL algorithm built on MuZero. By making three changes, namely using self-supervised learning to learn the environment model, predicting the value f, and using the model to correct off-policy value targets, EfficientZero achieves state-of-the-art performance on the Atari 100k benchmark. The authors also provide an efficient implementation for future research.

**Ethical Concerns:**

N.A.

**Limitations And Societal Impact:**

Yes.

**Main Review:**

The proposed approach is the first one to achieve human-level (or comparable) performance with limited data. The three modifications have been studied by ablations, and every of them is shown effective respectively. Besides, the paper is well structured and easy to follow. However, I still have the following comments that the authors need to address.

1. MuZero Reanalyze in [30] uses a target network to provide the n-step bootstrapped target value, while EfficientZero uses MCTS root values in off-policy correction. In addition, EfficientZero in this paper also uses dynamic n (denoted by l). The ablation tests in this paper should separate these two issues (using MCTS root values and dynamic n) for analysis. It is more interesting to see which is more significant. BTW, it is critical to show the required computation costs like Table 1 in [30].
2. In the experiments (Section 6.5 and Appendix), the ablation study of removing three components (the self-supervised consistency loss, the end-to-end value prefix learning, and the model-based off-policy correction) are applied to 3 games only in Section 6.5 and 5 other games in Appendix. It would be better if these can be applied to all the other games.
3. This paper does not do ablation tests for data augmentation, which has been shown to have non-trivial improvement of the performance in SPR and DrQ. However, it is still interesting to see ablation tests for EfficientZero (without data augmentation) to see how significant it is when compared with other components.
4. This paper states “It is worth to note that MuZero does not explicitly learn the environment model. Instead, it solely relies on the reward and value prediction to learn the model.” However, the policy signal should also be involved for learning the environment model.

Following are related to the presentation.
Line 33: Muzero -> MuZero
Line 74: reference of SimSiam should be [7]
Line 88: such as Go, Chess and Shogi [37, 37] -> [36, 37]
Line 148: To be precise, Formula (2) is called a variant of the PUCT algorithm.
Line 175: “Either” the reward “or” the value “provide” enough training signal -> “Neither” the reward “nor” the value “provides” enough training signal
Line 178: the predicted “reward” still have large prediction errors -> “rewards”
Figure 1: “next-state” component should be “dynamic” or “transition”
Line 229: an important “roll” in MCTS -> “role”
Line 267: “use l rewards” should be “use l-step rewards”

**Time Spent Reviewing:**

6 hours

---

> ### Author Response · Authors · 2021-08-10
> **Response to Reviewer CuAV**
>
> Thank you for your comments and corrections for the typos!
>
> For the first suggestion "It is more interesting to see which is more significant  (using MCTS root values and dynamic n). BTW, it is critical to show the required computation costs like Table 1 in [30]", here we add more ablation study about this. Specifically, we evaluate the performance of the EfficientZero model with only dynamic l and with only MCTS root value. The results are listed as follows:
>
> | Game         | Full        | w.o. off-policy correction | w.o. dynamic l | w.o. MCTS root value |
> | ------------ | ----------- | -------------------------- | -------------- | -------------------- |
> | Asterix      | 6218.8      | 2706.3                     | 3263.0         | **6288.0**           |
> | Breakout     | 388.8       | **468.6**                  | 427.0          | 387.8                |
> | Demon Attack | **10536.6** | 8640.6                     | 9211.1         | 10063.0              |
> | Gopher       | **2828.8**  | 2275.0                     | 2459.2         | 2651.0               |
> | Pong         | **19.8**    | 19.5                       | 19.2           | 14.5                 |
> | Qbert        | **15268.8** | 3948.4                     | 7945           | 14738.0              |
> | Seaquest     | **1321.0**  | 1248.0                     | 1292.0         | 876.0                |
> | UpNDown      | **10238.1** | 3240.0                     | 4772.0         | 9925.6               |
>
> As illustrated here, we find that the dynamics l is more significant in the limited data setting as the performance without dynamic l is poorer in more environments than that without MCTS root value. To sum up, the dynamic l is more important than the MCTS root value with limited data. We will show a computation cost for this in the final version.
>
> For the second comment "ablations of three components on all games", we agree that the ablation study of removing three components can be applied to all the other games. Due to the limited time during rebuttal, we will finish those ablation experiments and update that in the final version.
>
> For the next question about the ablation study of data augmentation, here we have done the experiments for the environments mentioned in the table and the results are listed as follows.
>
> | Game         | Full        | w.o. consistency | w.o. data augmentation |
> | ------------ | ----------- | ---------------- | ---------------------- |
> | Asterix      | 6218.8      | 1350.0           | **13884.0**            |
> | Breakout     | 388.8       | 12.0             | 365.2                  |
> | Demon Attack | **10536.6** | 5973.8           | 8730.0                 |
> | Gopher       | **2828.8**  | 1155.0           | 1823.75                |
> | Pong         | **19.8**    | -8.5             | 13.9                   |
> | Qbert        | **15268.8** | 2304.7           | 14286.0                |
> | Seaquest     | **1321.0**  | 460.0            | 1125.0                 |
> | UpNDown      | 10238.1     | 3040.0           | **16380.0**            |
>
> As shown here, we find that data augmentation can improve the performance of most environments but it will also drop the performance of few others, such as Asterix and UpNDown. Although the role of data augmentation is a little bit mixed, it still offers benefits for most of the environments. As the table above shows, it can improve the performance to some extent but is less important than the component like self-supervised consistency.
>
> For the last one, it is true that "the policy signal should also be involved for learning the environment model". It is a typo and we will update that in the final version.
>
> Finally, thanks for your text suggestions, and we will take them in the final version.

---

### Official Review · Reviewer_deNp · 2021-07-11

**Rating:** 7
**Confidence:** 4

**Summary:**

This paper presents EfficientZero - a variant of MuZero that targets data efficiency in the low-data regime. The authors identify 3 factors that hinder the data efficiency of MuZero in the low-data regime, namely lack of supervision on learning the model, compounding errors in value estimation, and lack of off-policy correction with stale data. To provide richer supervision for learning the model, EfficientZero employs an auxiliary self-supervised objective similar to SimSiam. To combat compounding errors, EfficientZero directly learns the "value prefix" rather than per-step rewards. Finally, EfficientZero utilizes the model to address off-policy correction. EfficientZero achieves a new state of the art on the Atari 100k benchmark, outperforming existing methods by a large margin. Ablation studies demonstrate the importance of all three proposed components - removing any of them results in significant performance drop.

**Limitations And Societal Impact:**

There is no foreseeable negative societal impact.

**Main Review:**

Things to like:
* 1. Overall the paper is well written. The authors successfully stated the problem of interest, their hypothesis on the potential issues with current methods, their proposed solutions, and the empirical verification. The writing is easy to follow. Although there are some minor errors and typos, they do not hinder the readability.
* 2. EfficientZero addresses important problems in value-equivalent models and model-based RL. MuZero, and other value-equivalent models, all suffer the problem of lacking learning signals. The compounding error is also a well-known issue in the model-based RL context. Admittedly, the solutions used by EfficientZero are not completely novel and are mostly variants of existing methods. However, the empirical success of applying them to state-of-the-art agents like MuZero is still significant. I believe the community can benefit from the success of EfficientZero.
* 3. The empirical results are very impressive. EfficientZero achieves a new state of the art on the Atari 100k benchmark and is the first to achieve super-human performance in terms of the mean human-normalized score. This is a big step forward.
* 4. An open-source implementation of MuZero/EfficientZero can be a valuable asset to the community.

Things that can be improved:
* 1. There is some inconsistency between the identified issues and the proposed solutions. In L40-42, it says "... three components are essential to the image input model-based RL agent: ..., ..., and an efficient use of the model to explore the world". I understand the third component as using the model for better exploration. However, the corresponding solution is to use the model for off-policy correction. Unless I misunderstand something, this seems a bit inconsistent to me. Similar confusion was found in L256 as well. "[predicting the value prefix] helps the MCTS to explore better". I do not fully understand the connection between less compounding error and better exploration.
* 2. Asadi *et al* [1] proposed a multi-step model to combat the compounding error, which also directly learns the "value prefix". The authors should as least mention it in the related work.
* 3. For model-based off-policy correction, MuZero already uses the value output of MCTS as the bootstrap target (see the text associated with Eq. (1) in [3]). Section 4.3 is not clear on this. What is new here is a heuristic for choosing the unroll length $l$.
* 4. The authors could include a bit more discussion on the relative importance of the three components in the ablation study section. Based on the results in Table 1 in the appendix, removing the consistency loss seems to cause the biggest performance drop. Does it imply that richer learning signals are what MuZero lacks the most in the low-data regime?

Questions for the authors:
* 1. Since off-policy correction is crucial in this low-data regime, did the authors try existing methods like Retrace($\lambda$) [2]? The current solution may be too aggressive at truncating the trajectories, which may cause slower learning.
* 2. Could the authors comment on the impact of the three components in the high-data regime, like 200M frames? The off-policy correction may be less effective in a more online setting as the problem itself is less severe. But the other two components may still help.

Minor Errors and Typos:
* 1. L22, a reference to AlphaZero is missing.
* 2. L33, "Muzero" --> "MuZero".
* 3. L88, duplicated reference [37].
* 4. L151, "... prioritize explore promising part of the tree" --> "prioritize exploring...".
* 5. L350-351, the term "self-play network" pops up from nowhere.

References:
* [1] Asadi *et al*, 2019, *Combating the Compounding-Error Problem with a Multi-step Model*.
* [2] Munos *et al*, 2016, *Safe and efficient off-policy reinforcement learning*.
* [3] Schrittwieser *et al*, 2020, *Mastering Atari, Go, Chess and Shogi by Planning with a Learned Model*.

**Time Spent Reviewing:**

4

---

> ### Author Response · Authors · 2021-08-10
> **Response to Reviewer deNp**
>
> Thank you for your comments and corrections for the typos!
>
> For the first suggestion "There is some inconsistency between the identified issues and the proposed solutions.", we feel grateful for this because it is true that such writing is not accurate enough. Here is the explanation: the less compounding error can indirectly result in better exploration, since during the policy rollouts, the exploratory action depends on the MCTS tree search results. The less compounding error, the more accurate the tree search would be, which will likely to results in better exploration. Although there are connections between the less compounding error and the better exploration, their relationship is relatively weak. We will improve the presentation in the final version.
>
> We agree with the reviewer's second and third suggestions. We will discuss the related work and clarify Section 4.3.
>
> For the last suggestion "a bit more discussion on the relative importance of the three components in the ablation study section. " We agree that the richer learning signals are the aspect Muzero lacks most in the low-data regime. After fixing the poor supervision issues, the other two components become indispensable to superior performance. We will include more discussions on the relative importance of the three components in the ablation study.
>
> For the first question on Retrace, we haven't tried Retrace yet, but that was on our to-do list and we are going to try it out soon. Hopefully, it can provide a better bias-var trade-off.
>
> For the second question on performance in the high-data regime such as 200M frames. We did find that even in the high data regime, the temporal consistency can significantly accelerate the training. The value prefix seems to be helpful during the early learning process, but not as much in the later stage. We are still trying to figure out why. We haven't tried the off-policy correction in the high data regime since that doesn't seem to be necessary and we also expect it would not be very helpful. Since we focus on the data-efficient regime, unfortunately, we don't have a rigorous number table for the 200M case.
>
> Finally, thanks for your text suggestions, and we will correct them in the final version.

---

> > ### Comment · Reviewer_deNp · 2021-08-31
> > **Thank you for the response**
> >
> > Thank you for the response! It answers my questions well. I will stick to my initial evaluation and vote for acceptance.

---

### Official Review · Reviewer_4xQE · 2021-07-16

**Rating:** 7
**Confidence:** 3

**Summary:**

The authors consider the problem of sample-efficient reinforcement learning focused on image-based data, particularly in the Atari domain. Building on the MuZero model, they introduce three key changes to improve the learned model and thereby the sample efficiency: (a) a loss to enforce consistency between rolled-out and direct state abstractions learned by the model; (b) an end-to-end approach for predicting the sum of discounted rewards ("value prefix") used to generate a MCTS Q-value estimate, and (c) the use of an additional MCTS rollout for bootstrapping the state-value estimate with replay buffer data, to limit the off-policy bias of multi-step estimates from old data. They show that the proposed changes result in a significant performance boost in the low-data (100k) Atari domain, over both the original muZero and a variety of sample-efficient baselines. The also provide an ablation study that demonstrates improvement from the combination of the three interventions over each individually. The authors also release the source code.

**Ethical Concerns:**

None.

**Limitations And Societal Impact:**

No issues.

**Main Review:**

**Quality:**

The interventions generally seem quite reasonable, well-motivated and applicable to other tasks, and the empirical results are compelling. The paper has a good related work section (including quite recent results like [9] that work on similar themes). The authors also release their source code (I assume the url will be included in the supplementary in the final submission).

One key point that was not clear to me is the computational cost of the MCTS bootstrapping in the off-policy correction. What is the difference in train time with and without this?

I'm not fully convinced by the self-consistency reconstructions on page 2 in the appendix. The decoder is trained independently to reconstruct the observations from the state abstractions s_t? The decoder (particularly in the middle row) appears to be reconstructing impossible states, whereas I would assume that in the failure state it would simply reconstruct a static image of a probable game state (independent of s_t).

Can you list MuZero as well in Table 2?

**Originality:**

The paper proposes several improvements on top of the existing MuZero architecture. The self-supervised consistency loss appears to have been concurrently proposed (though not yet peer-reviewed) by de Vries et al., and the imaginary rollouts have been used by others albeit without the fine-grained tradeoff with real-world experience. The "value prefix" learning seems to be the most original idea and is compelling; although it seems to be the weakest intervention according to the ablations, the authors do show an additional train vs. validation comparison to illustrate the effectiveness. Altogether, I believe that illustrating the individual and combined benefits of the interventions and releasing the code illustrates a strong contribution to the community.

**Significance:**

As mentioned, the gains in overall performance are impressive. It would be nice to see results in one other problem domain (e.g. 3D exploration); the focus on Atari in the RL community risks overfitting to a single domain, although the proposed interventions don't appear to be domain-specific.

**Clarity:**

The aleatory uncertainty point (Line 177) is used to motivate the value prefix loss. I found both this and the explanation in 4.2 to be somewhat confusing, although I see the point after some reflection. I think it would help to point out that the network might equally assign responsibility for a reward to multiple states in the future, leading to an additive error in the estimate.

Can you explain / expand on the role of the projector in the main text in Section 4.1?

Figure 4: why do you report two different losses in train and validation?

Minor text suggestions:

Line 109: ...and use[s] model-based value estimate[s] to bootstrap.

Line 135: Open bracket in \tilde{V}(s_{t+k}) is superscript

Equation 2: Can you add brackets around the argmax argument for clarity?

Line 175: [Ne]ither the reward ...

Line 182: Off-policy [nature] of multi-step value

Line 221: ...adjacent populations provide [natural] two views... (natural is confusing here)

Line 229: The state aliasing problem has [a] negative...

Figure 2 caption: ...the right player didn't move and miss[ed] the ball.

Line 296: ...since it requires [fewer] samples.

Line 303: ...run[s] MCTS search...

Line 304: ...into [a] CPU part and [a] GPU part...

Line 307: ...to avoid performance issue[s] with Python...

Line 331: ...which also first propose[d] the Atari...

Line 333: ...a method that use[s] contrastive...

Line 346: ...our limit[ed] data scenario...

Line 360: ...can outperform [a] human player....

Line 361: ...our method outperform[s] human[s] in...

Line 365: ...three issues that prevent[s] MuZero (remove s)

Line 373: Due to limited computation resources... (no on)

Line 361: cause the [difficulty] in predicting...

**Time Spent Reviewing:**

8

---

> ### Author Response · Authors · 2021-08-10
> **Response to Reviewer 4xQE**
>
> Thank you for your comments and corrections for the typos!
>
> For the first question "One key point that was not clear to me is the computational cost of the MCTS bootstrapping in the off-policy correction. What is the difference in train time with and without this?" The computation cost of off-policy correction is two times on the reanalyzed side. However, due to the parallel implementation, it does not slow down the training process. Here is a more detailed explanation. On the implementation side, the training process of EfficientZero is split into the reanalyzing part and the SGD part. The reanalyzing part is aimed at preparing target rewards/values/policies via reanalyzing the past trajectories with target models and MCTS. The SGD part is aimed to update the parameters of the models given the prepared target stuff. In EfficientZero, the reanalyzing part and the SGD part run in parallel and they communicate through a queue that contains the training target. The bottleneck of training speed is the SGD part. The MCTS bootstrapping is in the reanalyzed part and it will make the reanalyzing two times slower. However, it will not slow down the training (or SGD part) speed.
>
> As for the second one "The decoder is trained independently to reconstruct the observations from the state abstractions s_t?", the answer is yes.
>
> And for the question "I would assume that in the failure state it would simply reconstruct a static image of a probable game state (independent of s_t).", the reason for reconstructing impossible states is the distributional shift. During the decoder training time, it is only trained to reconstruct the latent state $s_t$ obtained from running the encoder on $o_t$. However, during test time, the decoder is not only used to decode latent state directly encoded from $o_t$, but also used to decode inferred latent state from the dynamics model. This distributional shift between the training and testing of the decoder causes the reconstruction to be some non-probable state. Although there are ways to improve the decoder training, here we aim to show the difference between the model with consistency loss versus the one without.
>
> For the next question "Can you explain/expand on the role of the projector in the main text in Section 4.1?": It is common in self-supervised training because the representations of the last layer are not the best because the representations from the last layer are too restricted to the pretext task, thus not general enough for the downstream task. Specifically, the second or the third layer from the last would be better. For example, MoCo[1], Byol[2], and Simsiam[3], which are well-known self-supervised algorithms, choose the second from the last layer as the representations for the downstream tasks. Therefore, we follow the implementations of the SimSiam[3].
>
> For the last one "Figure 4: why do you report two different losses in train and validation?", we display the cross-entropy loss in training because the training objective function is the CE loss, but for validation, the L1 loss can be more intuitive. However, we agree to report L1 loss both in training and validation for consistency. And we will update that in the final version.
>
> Finally, thanks for your text suggestions, and we will take them in the final version.
>
> [1] He, K., Fan, H., Wu, Y., Xie, S., & Girshick, R. (2020). Momentum contrast for unsupervised visual representation learning. In *Proceedings of the IEEE/CVF Conference on Computer Vision and Pattern Recognition* (pp. 9729-9738).
>
> [2] Grill, J. B., Strub, F., Altché, F., Tallec, C., Richemond, P. H., Buchatskaya, E., ... & Valko, M. (2020). Bootstrap your own latent: A new approach to self-supervised learning. *arXiv preprint arXiv:2006.07733*.
>
> [3] Chen, X., & He, K. (2021). Exploring simple siamese representation learning. In *Proceedings of the IEEE/CVF Conference on Computer Vision and Pattern Recognition* (pp. 15750-15758).

---

> > ### Comment · Reviewer_4xQE · 2021-08-17
> > **Response to author**
> >
> > Thank you, I am satisfied with the response and will maintain my score as-is.
> >
> > If accepted and there is sufficient room (or in the appendix), I would like to see the computational cost clarification and a sentence on the projector citing prior work.

---

### Official Review · Reviewer_gNMH · 2021-07-17

**Rating:** 5
**Confidence:** 4

**Summary:**

This paper presents an algorithm called EfficientZero, a sample efficient model-based RL algorithm built upon MuZero. The authors demonstrate the efficiency of the proposed method via atari games, and call for the research of MCTS based RL algorithms in a wider community. The core contribution lies in the adaption of MuZero to settings where data is limited. The paper is overall interesting, easy to follow, and technically sound.

**Limitations And Societal Impact:**

I would suggest the authors to provide a thorough discussion on the limitations of this work, compared to existing works.

**Main Review:**

Originality & significance:  I have concerns regarding the novelty and applicability. From the perspective of methods, we have seen the application of model based, model free, and planning to atari games. This paper is a direct enhancement of MuZero, I would like to see how sample efficient it is on the testbeds used in Muzero, e.g., board games, with limited data settings. Since the vision of this paper claims to contribute a step towards real world RL applications such as robot manipulation, I would like to see some discussions on how the proposed method would be a step forward since atari is still a toy game mostly used in academia. At least the authors are encouraged to conduct experiments on simulation environments (e.g., continuous action space), which is similar to what "Mastering Atari with Discrete World Models" did.
Also, MCTS based methods usually suffer from handling real time senarios, where  a significant real world application requires quick response,e.g., online gaming, auto driving, robot manipulation, etc. A discussion on this point is necessary, since the authors hope to accelerate applicability of mcts based reinforcement learning.

Quality: the paper is technically sound; the experiments design for atari is ok. But I would like to see more experiments on other cases with limited data. As claimed by the authors, EfficientZero is ought to be a sample efficient model based RL algorithm for handling cases with limited data.

Clarity: the paper is overall easy to follow, but with some grammatical issues. The authors need to improve the wording.

**Time Spent Reviewing:**

2

---

> ### Author Response · Authors · 2021-08-10
> **Response to Reviewer gNMH**
>
> Thank you for your comments!
>
> In RL algorithms, Atari is a widely used benchmark, and the improvement of the Atari games performance can transfer to other tasks in general. We did not claim the first to apply model-based methods on Atari, but we claim a new and general model-based algorithm that has superior performance, validated on the Atari benchmark. Therefore, although there are a good amount of works on Atari games, such improvement in our work is still remarkable in general.
>
> In terms of your concerns, we plan to apply EfficientZero to other environments like board games and robotics environments. However, due to the large computation requirement of board games and the limited time during the rebuttal period, we are not able to complete the study on them. As for the robotics environments with continuous action space, We admit that our work is limited to the environments with discrete action space. It is non-trivial to implement MCTS-based methods for the environments with continuous action space. Recently, Hubert et al. [1] have proposed Sampled MuZero, an extension of the MuZero, to deal with continuous control tasks through a sampling modification on MCTS. However, there are some crucial modifications that are not easy to follow with the limited rebuttal time. Therefore, here we try to apply EfficientZero to some simulated robotics environments on the DMControl100k benchmark through a trivial action discretization method. DMControl100k benchmark contains some common and continuous simulated robot manipulation environments with only 100k environment steps, which is widely used as a benchmark in the sample efficient algorithms. In our implementation, we discretize each continuous action dimension into 5 discrete actions and use reward scaling as what we do on the Atari100k benchmark. We compare EfficientZero with the previous SoTA CURL[2] and other competitive alternative methods such as Dreamer[3] and SAC-AE[4]. We also compare it with two baselines: Pixel SAC and State SAC, which apply SAC directly to pixels and ground truth low dimensional states respectively. The performance of State SAC is the oracle performance since it doesn't need to learn the states from images. The results are listed as follows.
>
> | 100K STEPS SCORES  | EfficientZero  | MuZero          | CURL         | DREAMER       | SAC-AE      | PIXEL SAC   | STATE SAC  |
> | ------------------ | -------------- | --------------- | ------------ | ------------- | ----------- | ----------- | ---------- |
> | CARTPOLE, SWINGUP  | **813$\pm$19** | 218.5 $\pm$ 122 | 582$\pm$ 146 | 326$\pm$27    | 311$\pm$11  | 419$\pm$40  | 835$\pm$22 |
> | REACHER, EASY      | **952$\pm$34** | 493 $\pm$ 145   | 538$\pm$ 233 | 314$\pm$155   | 274$\pm$14  | 145$\pm$30  | 746$\pm$25 |
> | BALL IN CUP, CATCH | **942$\pm$17** | 542 $\pm$ 270   | 769$\pm$ 43  | 246 $\pm$ 174 | 391$\pm$ 82 | 312$\pm$ 63 | 746$\pm$91 |
>
> Compared with CURL[2], the previous SoTA, and other methods, EfficientZero achieves the state-of-the-art results and even outperforms the state-SAC which consumes the states as input instead of images. MuZero is much poorer compared with EfficientZero. Therefore, EfficientZero is a sample efficient model-based RL algorithm for handling cases with limited data, not only Atari games. In the future, we plan to incorporate the Sampled Muzero [1] technique into our framework, to deal with continuous action space systematically.
>
> As for the limitations, we agree that "MCTS based methods usually suffer from handling real-time scenarios, where a significant real-world application requires quick response". Specifically, for better results, people will still take a long time search in the evaluations if time is not limited. However, in MuZero or EfficientZero, there exists a policy network.  Therefore, we can use the policy network instead without searching in real-time cases. We also agree that it is significant to speed up the MCTS in real-time scenarios. Actually, one of our future works is to speed up the MCTS inference so as to obtain a quicker response in applications.
>
> [1] Hubert, T., Schrittwieser, J., Antonoglou, I., Barekatain, M., Schmitt, S., & Silver, D. (2021). Learning and Planning in Complex Action Spaces. *arXiv preprint arXiv:2104.06303*.
>
> [2] Srinivas, A., Laskin, M., & Abbeel, P. (2020). Curl: Contrastive unsupervised representations for reinforcement learning. *arXiv preprint arXiv:2004.04136*.
>
> [3] Hafner D, Lillicrap T, Ba J, et al. Dream to control: Learning behaviors by latent imagination[J]. arXiv preprint arXiv:1912.01603, 2019.
>
> [4] Yarats D, Zhang A, Kostrikov I, et al. Improving sample efficiency in model-free reinforcement learning from images[J]. arXiv preprint arXiv:1910.01741, 2019.

---

### Decision · Program_Chairs · 2021-09-27

**Decision:**

Accept (Poster)

**Comment:**

The reviewers were in universal agreement the paper proposed a novel, well justified method with good empirical results. Please include results on mujoco; it would be particularly interesting to include the ablation where MCTS is only used to generate learning targets, but not to act, in order to improve response time (you suggest this ablation in the rebuttal, and is also investigated in [1]). Also, if possible, please discuss or include some experiments in the 'data-rich' regime (high number of training frames). A method that strongly improves performance in the low-data regime, but catastrophically degrades performance as the amount of experience scales up, is of more limited interest than one which is data efficient while retaining good performance in data-rich regime.

[1] On the role of planning in model-based deep RL, Hamrick et. al